# Human-assisted Robotic Policy Refinement via Action Preference Optimization

**Wenke Xia[1,3,4,*], Yichu Yang[2], Hongtao Wu[2], Xiao Ma[2], Tao Kong[2], Di Hu[1,3,4,†]**

[1] Gaoling School of Artificial Intelligence, Renmin University of China, Beijing
[2] ByteDance Seed
[3] Engineering Research Center of Next-Generation Intelligent Search and Recommendation, MOE
[4] Beijing Key Laboratory of Research on Large Models and Intelligent Governance

## Abstract

Establishing a reliable and iteratively refined robotic system is essential for deploying real-world applications. While Vision-Language-Action (VLA) models are widely recognized as the foundation model for such robotic deployment, their reliance on offline expert demonstrations critically limits their capacity for *post-deployment refinement*. To mitigate this limitation, we introduce **Action Preference Optimization (APO)**, a method designed to refine VLA models by human-assisted preference alignment gathered through interaction with environments. This method begins with a human-robot collaboration framework for reliable failure correction and interaction trajectory collection through human intervention. However, directly leveraging these interaction trajectories for preference optimization is non-trivial due to the challenges of irreversible robotic actions and token distribution mismatch. To solve this, APO proposes an adaptive reweighting algorithm with binary desirability signals derived from interaction, empowering VLA models effectively suppress failure-prone actions while enhancing corrective action adaptation. Ultimately, APO equips VLA models with the crucial capability to learn from failure, paving the way for their iterative refinement and reliable deployment in dynamic environments. The experiments conducted in simulation and real-world scenarios prove superior generalization and robustness of our human-assisted framework across a variety of manipulation tasks. We believe this work could bring insights for efficient and stable optimization of VLA models through human-robot collaboration. The code and dataset are released at https://github.com/GeWu-Lab/Action-Preference-Optimization.

## 1 Introduction

Fostering continuous improvement is crucial for the development of robust robotic manipulation systems in real-world scenarios [10, 28, 44]. Benefiting from the capacity for generalizable reasoning and scalable learning, Vision-Language-Action (VLA) models [3, 4, 8, 18, 43, 21, 47] have been widely recognized as the foundation model for such robotic deployment systems. However, prevailing training paradigm for these models hinges on large-scale, offline datasets of expert demonstrations. This severely limits their *post-deployment refinement*, as they lack the intrinsic ability to continually learn from failures or adapt to novel scenarios encountered in the real world.

To enhance the continuous learning ability of robotic systems, interactive imitation learning frameworks [7, 17] have been developed to refine error-prone trajectories via iterative human-in-the-loop

---

[*]Work is done during internship at ByteDance Seed
[†]Corresponding author

39th Conference on Neural Information Processing Systems (NeurIPS 2025).

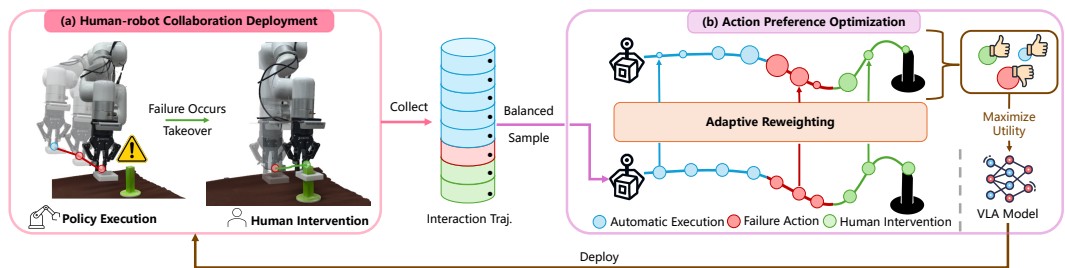

Figure 1: Our method consists of two key components: (a) the human-robot collaboration deployment framework for reliable deployment and interaction trajectory collection with human intervention. (b) the action preference optimization process with adaptive reweighting for VLA models learning from sub-optimal interaction trajectories. The size of each circle represents its weight during training.

correction feedback. Among them, behavior cloning [15, 23, 35] has been widely utilized to fine-tune a base policy model with manually corrected intervention data in a supervised learning fashion. In contrast, recent methods [20, 26] seek to propose effective off-policy reinforcement learning algorithms from sub-optimal human intervention trajectories. However, behavior cloning fails to fully exploit failure trajectories, which are valuable signals for learning robust policies. At the same time, reinforcement learning methods encounter significant scalability limitations in training large-scale VLA models, due to the inherent instability and challenge of developing generalizable value functions. To date, the effective adaptation of VLA models for downstream manipulation tasks remains understudied, particularly within sub-optimal human intervention paradigms.

To bridge this gap, we propose **Action Preference Optimization (APO)**, a new paradigm moves beyond the limitations of both behavior cloning and reinforcement learning. By learning from action-level preferences captured during interactions, our approach fully *exploits the valuable information in failure trajectories while maintaining the optimization stability* required for large-scale VLA models.

Our method is founded on a human-robot collaboration framework designed to ensure reliable deployment while simultaneously generating data for policy refinement, as illustrated in Figure 1(a). When the robot encounters challenging situations, real-time human interventions not only guarantee successful task completion but also provide corrective trajectories. These trajectories are collected as preference pairs to refine the policy. Furthermore, to address the imbalanced distribution of action types in the collected data, we employ a balanced sampling method. This ensures a proportional representation of all interaction data for the subsequent VLA preference optimization

However, directly applying this preference data to fine-tune autoregressive VLA models presents two significant challenges: (1) Irreversible interaction: While LLMs often require paired preference data, the irreversible nature of physical interaction makes it difficult to gather perfect positive-negative action samples under identical conditions. (2) Token probability mismatch: Autoregressive VLA models discretize continuous actions into tokens, causing a mismatch between token probabilities and the true action loss, which complicates preference alignment. To address these problems, we first employ Kahneman & Tversky's prospect theory [11, 40] to formulate a preference alignment objective that learns from binary desirability signals derived from interaction. This objective relaxes the demands of preference pairs, making it suitable for learning from irreversible robotic interaction trajectories. Furthermore, we propose an adaptive reweighting method that leverages decoded continuous actions to guide preference optimization in the discrete action token space. This approach addresses the challenge of action token probability mismatch via the dynamic modulation of sample-wise training weighting, thereby concentrating gradient optimization on failure-prone interaction actions. Through weight refinement, we apply preference alignment optimization to VLA models, enhancing performance when *learning from sub-optimal manipulation correction trajectories*.

To systematically evaluate the effectiveness of our proposed system, we conduct a comprehensive set of experiments in RoboMimic [29] simulation environments. The empirical results demonstrate that APO facilitates rapid adaptation in in-distribution scenarios while maintaining robust performance across a variety of unseen perturbations. Furthermore, lifelong learning experiments demonstrate the framework's capacity for iterative improvement through human intervention. To evaluate the practical viability of the proposed framework, we conducted real-world experiments on fine-grained insertion tasks under a range of disruption conditions, demonstrating its robustness and applicability in real-world robotic manipulation scenarios.

## 2 Related Works

### 2.1 Vision-Language-Action Models

Achieving generalizable robotic manipulation remains a significant challenge within the field of robotics. Motivated by recent advances in foundation models [19, 39, 42, 32, 45, 48], some works [6, 12, 41] attempt to construct large-scale real-world robotic datasets to facilitate development of generalizable Vision-Language Action (VLA) models. Building upon these datasets, recent research [5, 18, 33] formulates robotic action prediction as a next token prediction problem within the framework of VLMs. In contrast, alternative studies [24, 38] investigate the applicability of diffusion-based methods to model multi-modal action distributions, thereby facilitating robustness in manipulation tasks. While these works focus on behavior cloning from expert demonstrations, Grape [49] proposes a trajectory-level preference alignment method to boost generalizability by incorporating both successful and failed trials. However, the requirements of paired trajectories under the same conditions make it infeasible in real-world scenarios. In this work, we propose the action preference optimization method to continuously refine VLA models by integrating human-in-the-loop intervention preference data.

### 2.2 Preference Alignment of Large Language Models

Contemporary methods [9, 30, 31, 50] implement Reinforcement Learning from Human Feedback (RLHF) through a two-stage method, which first trains a reward estimation model and optimizes LLMs to maximize the given estimated reward with a reinforcement learning method [13, 36, 25]. However, this paradigm is slow and unstable in practice. DPO [34] proposes a single-step alternative that reparameterizes the RLHF objective into a closed-form loss function to directly maximize the log-likelihood margin between preferred and dispreferred outputs. Extending this framework, KTO [11] introduces the human-aware losses for learning from a binary signal of whether an output is desirable, which bypasses the need for intricate preference annotation altogether. In this work, we adapt the preference alignment optimization method for Vision-Language-Action models. Through an adaptive reweighting approach, we mitigate the irreversible interactions and token probability mismatch challenges when transferring preference learning methods from LLMs to VLA models.

### 2.3 Human-robot Interactive Learning

Interactive imitation learning [1, 7] has been proposed to refine robot actions through human feedback. While prior research [17, 23, 35] necessitates constant human supervision to intervene in the robot's actions, more recent studies [14, 22, 46] have introduced dynamic models for automatic failure detection and real-time monitoring. In contrast, RLIF [26] leverages human intervention signals as rewards for off-policy RL, while HIL-SERL [27] presents a human-in-the-loop, vision-based RL system tailored for dexterous manipulation tasks. However, these RL approaches encounter difficulties in large-scale VLA model training, primarily due to unstable gradient optimization. In this work, we propose the action preference optimization method to ensure the stable optimization of policies from action-level preferences captured during interaction.

## 3 Method

In this section, we introduce Action Preference Optimization (APO), a method designed to facilitate continuous iterative improvement of Vision-Language-Action (VLA) models. As detailed in Algorithm 1, APO aligns the model with human preferences gathered through human-robot collaboration deployment within environment.

### 3.1 Human-robot Collaboration Deployment

To ensure reliable deployment and interaction trajectory collection, our method is founded on the human-robot collaboration framework for real-time intervention and interaction data acquisition.

We first collect an expert demonstration dataset $\mathcal{D}_e = \{\tau_e^i\}_{i=1}^{i=N}$, where each trajectory $\tau_e^i$ consists of observation-action pairs with expert annotations: $\tau_e^i = \{(o_t^i, a_t^i, c_t^i)\}_{t=1}^{t=T}$, where $c_t^i = 1$ indicates that $a_t^i$ is executed by human expert. We employ behavior cloning to fine-tune the pretrained VLA model

on these expert demonstrations, obtaining an initial base policy $\pi_\theta^0$. This policy is then deployed for interaction trajectory collection.

During policy execution, the human operator monitors policy execution and intervenes when the policy encounters challenging scenarios. Through this process, we could collect a set of interaction trajectories $\mathcal{D}_h = \{\tau_h^i\}_{i=i}^{i=M}$, where $c_t^i = 2$ represents the action is corrected by human intervention while $c_t^i = 1$ denotes the action is executed by policy. Further, we re-label the interaction trajectories to categorize the actions taken in the $K$ steps preceding human interventions as undesirable, annotated with $c_t^i = 0$. For each trajectory, we discretize the continuous action $a$ into discrete action token $\hat{a}$. Finally, we combine the expert demonstrations $\mathcal{D}_e$ and the interaction dataset $\mathcal{D}_h$ for further robotic action preference optimization.

## 3.2 Action Preference Optimization

To maximize the utility of sub-optimal interaction trajectories and ensure stable fine-tuning of the VLA model, we adopt the preference alignment optimization method to guide the model to learn from corrections and avoid failures.

Although previous Reinforcement Learning with Human Feedback (RLHF) methods [2, 34] have proven effective in LLM fine-tuning, there are additional challenges for the VLA models preference optimization in robotic manipulation:

- The irreversible robotic manipulation process makes it challenging to acquire meaningful paired positive-negative actions under the same observational conditions.
- The mapping of continuous robotic actions to discrete tokens by autoregressive VLAs causes a mismatch between token probability and continuous action errors, complicating preference optimization in action token prediction.

To address these issues, we adopt Kahneman & Tversky's prospect theory [40] for preference alignment optimization with binary desirability signals and propose an adaptive reweighting method to bridge the gap between discrete token prediction and continuous action regression. We first estimate the reward function $r_\theta$ of our model $\pi_\theta$ as standard approach [31, 34, 37]:

$$r_\theta(o, \hat{a}) = \log \frac{\pi_\theta(\hat{a}|o)}{\pi_{\text{ref}}(\hat{a}|o)}, \tag{1}$$

where $\hat{a}$ is the discrete action token and the reference model $\pi_{ref}$ is the base model $\pi_\theta^i$ at the beginning of each deployment-optimization loop shown in algorithm 1. Following [11, 40], we formulate the utility function $v$ as below to estimate the relative gain on the robotic data:

$$v(o, \hat{a}) = \begin{cases} \lambda_D \sigma\left(r_\theta(o, \hat{a}) - z_0\right) & \text{if } \hat{a} \sim \hat{a}_{\text{desirable}} \\ \lambda_U \sigma\left(z_0 - r_\theta(o, \hat{a})\right) & \text{if } \hat{a} \sim \hat{a}_{\text{undesirable}}, \end{cases} \tag{2}$$

where $\lambda_D$ and $\lambda_U$ are utilized for importance sampling, the $\sigma$ is the sigmoid function. To ensure that the model $\pi_\theta$ does not deviate excessively from the reference model $\pi_{ref}$, a penalty term $z_0$ is introduced. This term is defined as the KL-Divergence between $\pi_\theta$ and $\pi_{ref}$: $z_0 = KL(\pi_\theta||\pi_{ref})$. Incorporating $z_0$ into the optimization process guides the model to learn from preference pair data while simultaneously preserving knowledge acquired from prior models. We employ the following loss function $L$ to optimize the model $\pi_\theta$ using preference optimization with desirability signals:

$$L(\pi_\theta, \pi_{ref}) = \mathbb{E}_{x,y \sim D^h}[-v(x, y)]. \tag{3}$$

By minimizing the loss function, we aim for the model $\pi_\theta$ to get higher rewards for desirable pairs while avoiding predicting undesirable actions, in comparison to the reference model $\pi_{ref}$.

However, directly applying the preference alignment optimization from LLMs to autoregressive VLA models is problematic, primarily due to the differences in their respective token definitions. While word tokens correspond to distinct subwords, action tokens necessitate a non-differentiable mapping to continuous ground truth actions. This creates a discrepancy between the token classification probabilities and the regression loss associated with the continuous robotic actions.

To bridge the gap between token classification and continuous action regression in autoregressive VLA models, we introduce an adaptive reweighting method. This approach guides the model to

prioritize samples exhibiting large regression errors by first estimating the L1 loss of the continuous action $l$ for each sample, followed by batch-level normalization as detailed below:

$$w_i = \frac{l_i}{\sum_{i=1}^{i=B} l_i}. \tag{4}$$

The normalized weighting scheme operates by: 1) for desirable data, increasing the weight of samples with high action prediction errors, and 2) for undesirable data, increasing the weight of samples whose actions are proximate to the failure actions. By adaptively adjusting the values of $\lambda_D$ and $\lambda_U$ in Equation 1 using the normalized weights, we gain fine-grained control over the relative influence of each sample during training:

$$\lambda_D = 1 - e^{-\beta_D * w}, \tag{5}$$

$$\lambda_U = e^{-\beta_U * w}. \tag{6}$$

By incorporating preference alignment optimization via sample-wise weight refinement, we enhance the performance and optimization stability of the VLA model, when learning from sub-optimal manipulation correction trajectories.

In conclusion, we propose the action preference optimization method as demonstrated in algorithm 1. This approach leverages the human-robot collaboration deployment for reliable task execution and interaction trajectories collection, while the action preference optimization process provides stable autoregressive VLA optimization with adaptive reweighting. Through iterative human-robot collaboration deployment and action preference optimization, we could achieve continual improvement from interaction with environments for autoregressive VLA models.

---

**Algorithm 1** Action Preference Optimization

---

1: **Notations:**
2: $\mathcal{D}_e$: expert demonstrations, $\mathcal{D}_h$: interaction dataset, $\pi$: interaction policy
3: **Warm-start phase**
4: Collect $\mathcal{D}_e \leftarrow \{\tau_1^e, \ldots, \tau_N^e\}$
5: Initialize BC policy $\pi_\theta^0$
6: $\theta^* \leftarrow \arg\max_\theta \mathbb{E}_{(o,a)\sim\mathcal{D}^e} \left[\log \pi_\theta^0(a|o)\right]$
7: $\mathcal{D}_h^0 \leftarrow \mathcal{D}_e$
8: **Deployment-optimization loop**
9: **for** $i \leftarrow 0$ **to** $X$ **do**
$\quad \pi_{ref} \leftarrow \pi_\theta^i$
$\quad D_h^{i+1} \leftarrow \text{DEPLOYMENT}(\pi_\theta^i, \mathcal{D}_h^i)$
$\quad \pi_\theta^{i+1} \leftarrow \text{OPTIMIZATION}(\pi_\theta^i, \pi_{ref}, \mathcal{D}_h^{i+1})$
10: **end for**

---

11: **function** DEPLOYMENT($\pi_\theta, \mathcal{D}_h$)
12:     **for** n interaction rollouts **do**
13:         **while** task does not succeed **do**
14:             **if** human intervenes **then**
15:                 $a_t \leftarrow human, c_t \leftarrow 2$
16:                 $c_{t-K:t-1} \leftarrow 0$
17:             **else**
18:                 $a_t \leftarrow \pi_\theta(o_t), c_t \leftarrow 1$
19:             **end if**
20:             $\tau_i^h \leftarrow \tau_i^h \cup (o_t, a_t, c_t)$
21:         **end while**
22:         $\mathcal{D}_h \leftarrow \mathcal{D}_h \cup \tau_i^h$
23:     **end for**
24:     **return** $\mathcal{D}_h$
25: **end function**

26: **function** OPTIMIZATION($\pi_\theta, \pi_{ref}, \mathcal{D}_h$)
27:     **for** n gradient steps **do**
28:         Balanced Sample $(o_i, a_i, c_i)_{i=1}^{i=B} \sim \mathcal{D}_h$
29:         $l_i \leftarrow |\pi_\theta(o_i) - a_i|_1$
30:         $w_i \leftarrow \frac{l_i}{\sum_{i=1}^{i=B} l_i}$
31:         **if** $c_i \neq 0$ **then**
$\quad\quad\quad\quad \lambda_{D_i} = 1 - e^{-\beta_D * w_i}$
32:         **else**
$\quad\quad\quad\quad \lambda_{U_i} = e^{-\beta_U * w_i}$
33:         **end if**
34:         $\theta^* = \arg\max_\theta \mathbb{E}[-v(o, a, \pi_\theta, \pi_{ref}, \lambda_D, \lambda_U)]$
35:     **end for**
36:     **return** $\pi_{\theta*}$
37: **end function**

---

# 4 Experiments

To comprehensively evaluate our action preference optimization method for effective downstream adaptation, we propose experiments to validate the following questions:

- How effective is APO at promoting adaptation to in-distribution scenarios? Section 4.2
- Does APO maintain effective learning performance in novel scenarios despite various disruptions? Section 4.3
- Does APO demonstrate the ability to achieve iterative improvement during deployment? Section 4.4
- How well does APO generalize to different VLA models? Scetion 4.5
- Could APO be applied in fine-grained real-world scenarios? Section 4.6
- To what extent does the action-level preference optimization method enable APO to learn action correction? Section 4.7

## 4.1 Experiment Settings

**Implementation Details.** In this work, we fine-tune the OpenVLA [41] model for target manipulation tasks as the base model. We employ LoRA [16] for parameter-efficient tuning, configuring rank $r = 32$ with a batch size of 16 across 8 NVIDIA A100 GPUs. Further, we deploy the base model to interact with environments, where human operators perform real-time corrective interventions via a SpaceMouse device to rectify failures during execution. We set $K = 10$ to identify and annotate undesirable behaviors automatically. The human-assisted interaction trajectory is shown in Figure 1, which is segmented into robotic automatic execution, the failure action, and the human intervention types by the timing of human correction. Based on the interaction trajectories collected during task execution, we fine-tune the base model $\pi_{ref}$ with our action preference optimization method, using a learning rate of $5e$-$5$ and a batch size of 8 across 4 NVIDIA A100 GPUs. To ensure the stability of preference alignment training, we employ balanced sampling to ensure that each batch contains 50% expert actions, 25% human intervention actions, and 25% failure actions.

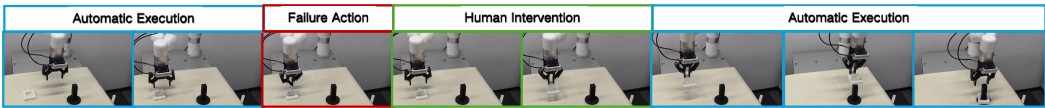

Figure 2: The demonstration of our human-assisted interaction trajectory.

**Simulation Environments Details.** For a comprehensive evaluation, we validate these methods on fine-grained manipulation tasks within the RoboMimic [29] simulation environment, such as 'make coffee' and 'toy assembly'. In the RoboMimic environment, we fine-tune the pretrained OpenVLA model for these 4 long-horizon manipulation tasks with 300 expert demonstrations. To optimize policy with human preferences, we collect 50 trajectories per task under different seeds for RoboMimic tasks. For evaluation, we conduct 50 trials under three unseen seeds for each task, and report the average success rate.

## 4.2 Comparison Experiments

We compare APO with other approaches to evaluate the effectiveness for VLA model fine-tuning. To ensure fairness, we fine-tune OpenVLA [18] for manipulation tasks as a base model and improve the base model with other comparison methods.

- **Dagger [35]:** We mix the expert demonstrations with interaction trajectories, fine-tuning the base model using a behavior cloning objective.
- **Sirius [23]:** we apply sample reweighting to prioritize human intervention data and fine-tune the base model using a weighted behavior cloning loss.
- **DPO [34]:** We generate paired negative samples for interaction trajectories by perturbing the actions predicted from the base model with Gaussian noise, and fine-tune the base model using these paired data with the DPO method.

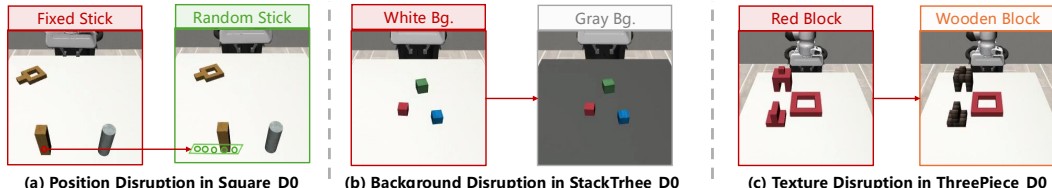

**(a) Position Disruption in Square_D0**  **(b) Background Disruption in StackTrhee_D0**  **(c) Texture Disruption in ThreePiece_D0**

Figure 3: In the position disruption setting, we change the position of the stick from a fixed point ○ to a random position from the rectangle ☐ in the Square_D0 task as illustrated in (a). In the background disruption setting, we replace the background with the gray one in the StackThree_D0 task as shown in (b). In the texture disruption setting, we replace the red blocks with the wooden ones.

- **TPO [49]:** We select positive and negative samples in interaction trajectories based on the timing of intervention, then fine-tune model with trajectory-wise preference optimization.

- **KTO [11]:** We select positive and negative samples in interaction trajectories based on the timing of intervention. Then, we sample positive and negative trajectories and optimize the base model with KTO, with the constraint $z_0 = KL(\pi_\theta || \pi_{ref})$.

As shown in Table 1, we first compare the behavior cloning objective methods. The results reveal that after fine-tuning with interaction data, these methods fail to outperform the base model, which demonstrates that existing behavior cloning approaches struggle to achieve efficient adaptation in the context of large-scale VLA models. A key challenge stems from the distribution shift between expert trajectories and interaction trajectories. Without mechanisms to retain the base model's knowledge under the under standard behavior cloning objectives, this shift makes it particularly difficult for large-scale VLA models to effectively fit the complex, multimodal distribution arising from the combined expert and interaction datasets.

We further provide results of the preference optimization based methods. By integrating a regularization constraint with the reference model $\pi_{ref}$, these methods could maintain useful knowledge from the reference model while achieving improvement from interaction trajectories.

Among all compared preference learning based methods, DPO yields the weakest performance. This result stems from its exclusive reliance on synthetic paired failure actions for optimization, which lacks exposure to real-world errors essential for teaching robots mistake avoidance through interaction. On the other side, TPO fails to deliver stable performance gains on multiple tasks while APO attains stable performance gains relative to the base model. The TPO method employs negative samples to regularize model preference alignment optimization, but introduces instability through random sampling. In contrast, APO utilizes KL divergence to estimate the mean margin between the updated model and the reference model, which not only enables more stable learning but also better preserves prior knowledge. Compared with KTO, APO leverages the adaptive reweighting method to achieve more precise control over the importance weights of both positive and negative samples, delivering more notable performance improvements.

Table 1: Comparison experiment results across 4 manipulation tasks in RoboMimic Simulation. The results demonstrate that our adaptive reweighting preference optimization method achieves stable improvement compared with other behavior cloning and preference optimization methods.

| Methods | Coffee_D0 | StackThree_D0 | ThreePieceAssebly_D0 | Square_D0 | Mean |
|---|---|---|---|---|---|
| Base policy | 44% | 46% | 44% | 28% | 40.5% |
| Dagger [35] | 42% | 50% | 36% | 28% | 39.0% |
| Sirius [23] | 34% | 52% | 34% | 38% | 39.5% |
| DPO [34] | 52% | 46% | 28% | 22% | 37.0% |
| TPO [49] | 54% | **54%** | 40% | 18% | 41.5% |
| KTO[11] | 48% | 52% | **46%** | **32%** | 43.5% |
| APO | **60%** | **54%** | **46%** | **32%** | **48.0%** |

Table 2: The results on disruption scenarios.

| Methods | Pos Dis. | Bg Dis. | Tex Dis. | Mean |
|---|---|---|---|---|
| Base policy | 12% | 42% | 10% | 21.3% |
| Dagger | 18% | **46%** | 4% | 22.7% |
| Sirius | 12% | 42% | 2% | 18.7% |
| DPO | 14% | 26% | 2% | 14.0% |
| TPO | 18% | 32% | 8% | 19.3% |
| KTO | 20% | **46%** | 6% | 24.0% |
| APO | **26%** | **46%** | **12%** | **28.0%** |

Table 3: The results on original tasks.

| Methods | Square | StackThree | ThreePiece | Mean |
|---|---|---|---|---|
| Base policy | 28% | 46% | 44% | 39.3% |
| Dagger | 16% | 46% | 30% | 30.7% |
| Sirius | 18% | 48% | 18% | 28% |
| DPO | 20% | 50% | 30% | 33.3% |
| TPO | 30% | 36% | 40% | 35.3% |
| KTO | 30% | 46% | **42%** | 39.3% |
| APO | **34%** | **62%** | 40% | **45.3%** |

(a) Coffee_D0 Success Rate    (b) StackThree_D0 Success Rate    (c) Human Intervention Frequency

Figure 4: Lifelong learning results of APO method.

## 4.3 Generalization to Novel Tasks

In this section, we assess APO's generalization capability under three novel scenarios, as illustrated in Figure 3. (1) **Position Disruption**: For the Square_D0 task, we replace the fixed initial stick position with randomized placements within a bounded operational area. (2) **Background Disruption**: In the StackThree_D0 task, we substitute the default white background with a gray one. (3) **Texture Disruption**: In the ThreePiece_D0 task, the original red blocks are transmuted to wood-grain visual properties. These experiments systematically evaluate robustness against spatial, background, and visual texture variations. To fine-tune the base model on novel disruption scenarios, we collect 20 interaction trajectories under disruption scenarios and combine them with 20 expert demonstrations from the original task for subsequent fine-tuning.

Our objective is to develop an action preference optimization method that facilitates continuous improvement, enabling performance enhancements in novel disruption scenarios while retaining original task capabilities during model fine-tuning. Thus, we evaluate the performance of the fine-tuned model across both disruption scenarios and original scenarios.

As shown in Tabel 2, the base policy exhibits some degree of performance degradation in disruption scenarios. However, the performance decline is relatively minor in cases of background disruption, whereas disruptions in object texture and position significantly impact performance. Both behavior cloning methods and preference optimization methods struggle to achieve significant performance improvements in novel disruption scenarios. In contrast, APO can effectively adapt to new disruption scenarios through adaptive reweighting.

Table 3 presents the performance of the optimized model after being fine-tuned from disruption data on the original task. The results reveal that behavior cloning methods exhibit severe catastrophic forgetting, resulting in substantial performance degradation. By contrast, the preference optimization method achieves mitigated performance decline with the constraints of the reference model.

Besides, APO utilizes adaptive reweighting to effectively integrate knowledge from both expert demonstrations and interaction trajectories. This mechanism not only facilitates learning from diverse data sources but also leads to improved performance on the original task.

## 4.4 The Performance of Lifelong Learning

To investigate whether APO can iteratively improve via environment interaction, we deploy APO to interact with environments while updating the model every 20 interaction rollouts. e provide comparison results using a behavior cloning policy trained with the same number of expert demonstrations as our baseline. For each updated model, we conduct 50 trials and report the success rate.

| Table 4: The results on $\pi$0-FAST model. | | | | Table 5: The results on real-world experiments. | | | | |
|---|---|---|---|---|---|---|---|---|

| Methods | Coffee_D0 | StackThree_D0 | Insert Square | Methods | In Dis. | Pos Dis. | Bg Dis. | Tex Dis. |
|---|---|---|---|---|---|---|---|---|
| Base policy | 68% | 64% | 85% | Base policy | 65% | 25% | 10% | 25% |
| Dagger | 64% | 66% | 85% | Dagger | 65% | 10% | 10% | 25% |
| TPO | 48% | 52% | 90% | TPO | 75% | 40% | 20% | 45% |
| APO | **76%** | **74%** | **95%** | APO | **85%** | **55%** | **30%** | **55%** |

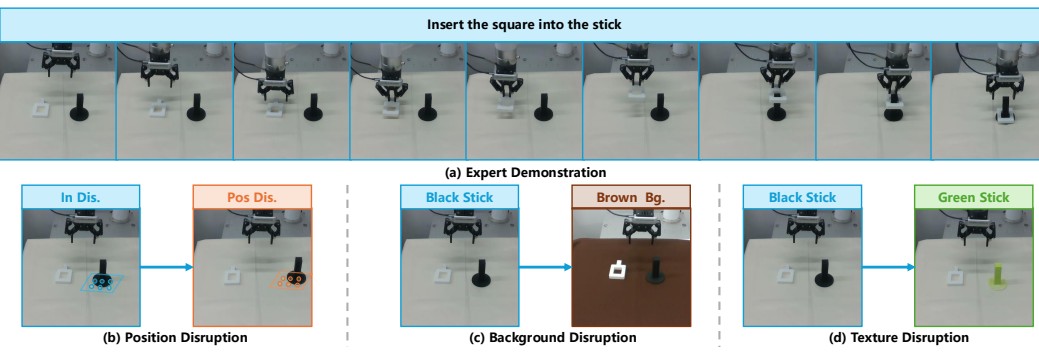

Figure 5: Demonstrations of real-world experiments with disruption settings.

As shown in Figure 4(a-b), APO achieves superior performance compared to the baseline, demonstrating its ability to effectively leverage sub-optimal human intervention trajectories for iterative model improvement. When the base policy exhibits diminishing improvement with increasing expert demonstrations, APO enables continual performance gains from the interaction trajectories. Besides, this improvement trend is accompanied by a corresponding reduction in the required human intervention ratio, as shown in Figure 4(c)

### 4.5 Generalization to various VLA models

To validate that APO can be adapted to different VLA models, we applied APO to fine-tune the $\pi$0-FAST [33] model. $\pi$0-FAST applies discrete cosine transform encoding to encode the action chunking into discrete tokens for VLA training. To adopt this model for downstream tasks, We regenerate the action tokenizer with 5 action chunking step for each task.

As shown in Table 1, the base model could achieve a higher success rate compared with the OpenVLA model, benefiting from its ability to predict action chunking for robotic manipulation. Further, we compare APO with both the behavior cloning method and preference optimization method, the results demonstrate that APO could achieve consistent improvement for the $\pi$0-FAST fine-tuning. The results prove that APO could be applied to the fine-tuning of various VLA models, achieving consistent performance gains.

### 4.6 Real-world Experiments

In this work, we conduct the challenging fine-grained robotic manipulation task "Insert the square into the stick" as shown in Figure 2(a), which requires the robot to grasp the square and precisely insert into the stick. To collect expert demonstrations, we utilize the spacemouse device to gather 100 high-quality trajectories at an action frequency of 20 *Hz*. We fine-tune the OpenVLA model with the collected demonstration as the base model. Further, we deploy the base model to interact with environments and propose the real-time human-in-the-loop interventions to collect 20 interaction trajectories for subsequent action preference optimization. All methods are evaluated under the same experimental setup, and we report the average success rate from 20 trials. For a comprehensive evaluation in real-world scenarios, APO was also tested on the "hang cup on the rack" and "put lemon on the plate" tasks. The corresponding results are detailed in the supplementary material.

To comprehensively evaluate APO, experiments are conducted not only under in-distribution but also across three distinct disruption settings as shown in Figure 2(b-e): (1) **Position Disruption:** We change the position distribution of the stick. (2) **Background Disruption:** We replace the tablecloth from white to brown. (3) **Texture Disruption:** We replace the black stick to the green one.

As demonstrated in Table 5, APO demonstrated robust adaptability to these downstream disruption scenarios. The results empirically validate the method's practical utility for real-world deployment in unstructured environments.

We also adopt APO to fine-tune the $\pi_0$-FAST model in the real world scenario. The results in Table 1 prove APO could achieve consistent performance gains over other methods.

### 4.7 Correction from Failure scenarios

In this work, we propose the APO method that enables models not only to avoid failure modes but also to self-correct within failure scenarios. As shown in Figure 6, we provide examples of failure correction across multiple tasks, demonstrating the corrective strategies learned by APO.

For instance, in Figure 6(a,b), when the model initially fails to grasp an object, APO identifies the failure and initiates a re-grasp attempt. Similarly, in Figure 6(c), when a precise insertion operation is obstructed, APO learns to iteratively adjust its gripper position until the insertion is successfully completed. These examples illustrate that APO has successfully learned to recover from common failure scenarios, which directly contributes to its improved overall performance.

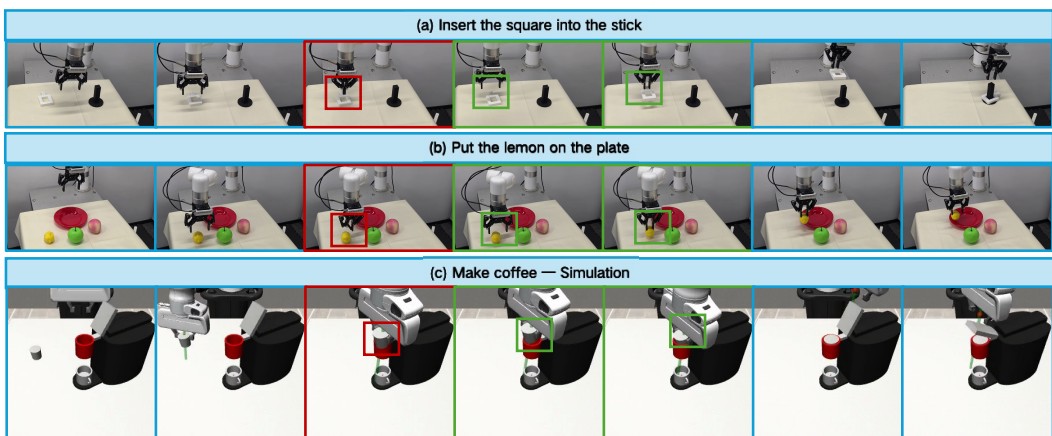

Figure 6: The rollout trajectory of APO. As indicated by the bold red and green boxes, APO can autonomously correct form failure scenarios.

## 5 Conclusion

In this work, we introduce the Action Preference Optimization (APO) method to fully exploit valuable information in failure trajectories while maintaining the stability required for large-scale VLA models. This method builds on a human-robot collaboration framework for reliable deployment, and utilizes an adaptive reweighting preference optimization algorithm with action-level binary desirability signals for stable VLA model optimization. Through APO, we could promote continuous improvement during the deployment of VLA models. We hope APO could bring insights for efficient and effective VLA model adaptation on downstream manipulation tasks.

**Discussion and Future Work.** While our work study the preference alignment optimization for VLA models, the experiments are based solely on autoregressive VLA models. Future work should explore a broader range of VLA frameworks, including regression-based approaches and diffusion policy models, to ensure the generalizability of our method across different architectures.

# 6 Acknowledgement

The project was supported by the fund for building world-class universities (disciplines) of Renmin University of China, and CCF-Zhipu.AI Large Model Innovation Fund. The project was also supported by National Natural Science Foundation of China (NO.62106272).

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

# SUPPLEMENTARY MATERIAL

## 1 Supplementary Video

In this work, we propose the action preference optimization method to correct interaction failure and achieve stable optimization for VLA models. In the supplementary video, we illustrate our human-assisted interaction trajectories collection process as demonstrated in Figure 1. We also provide comparison videos against other methods, highlighting the effectiveness of our approach in both real-world and simulation scenarios.

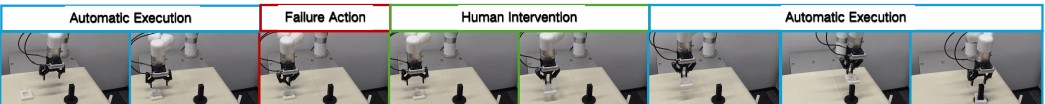

Figure 1: The demonstration of our human-assisted interaction trajectory.

## 2 Human-assisted Collaboration Deployment

In this work, we propose a human-assisted collaboration deployment framework to support reliable deployment and interaction trajectory collection. The blue block in Figure 1 illustrates the initial deployment of the base policy for autonomous environment interaction. However, the base policy is trained solely on expert demonstrations. When its predicted action causes failures, this model struggles to recover from these failure states, as shown in the red block. To address this, we provide human intervention to manually adjust the robotic arm's movements for failure correction, as shown in the green blocks.

Through this human-assisted approach, we ensure reliable deployment of the model in manipulation tasks. Furthermore, we annotate these interaction trajectories for subsequent preference learning. Specifically, we designate the last 10 actions before human intervention as undesirable data (representing failure actions), while the remaining trajectories serve as desirable data.

## 3 Implementation Details

In our work, we build the utility function $v$ as below to estimate the relative gain on the robotic data:

$$v(o, \hat{a}) = \begin{cases} \lambda_D \sigma \left( r_\theta(o, \hat{a}) - z_0 \right) & \text{if } \hat{a} \sim \hat{a}_{\text{desirable}} \\ \lambda_U \sigma \left( z_0 - r_\theta(o, \hat{a}) \right) & \text{if } \hat{a} \sim \hat{a}_{\text{undesirable}}, \end{cases} \tag{1}$$

where $z_0 = KL(\pi_\theta || \pi_{ref})$ to guide the model to learn from preference pair data while simultaneously preserving knowledge acquired from prior models. We compute the KL-divergence $z_0$ by leveraging the KTO [11] method, which leverages mismatched sample pairs for KL estimation. Further, we ignore the reject reward of the gripper action token to prevent erroneous rejection of the same gripper state.

## 4 More Real-world Experiments

### 4.1 Generalization to various VLA models

In this section, we adopt our method to fine-tune the $\pi_0$-FAST model. As shown in Table 1, the $\pi_0$-FAST model achieves a higher success rate, benefiting from its action chunking prediction. Besides, our method could achieve consistent performance gains in real-world experiments. Because our method needs to decode to continuous action for adaptive reweighting, however, the $\pi_0$-FAST model may fail to decode predicted action tokens into meaningful continuous actions, thus when the predicted action token sequences cannot be decoded to x, we would set the weight as 1 to promote the model focus on predicting correct action token sequences.

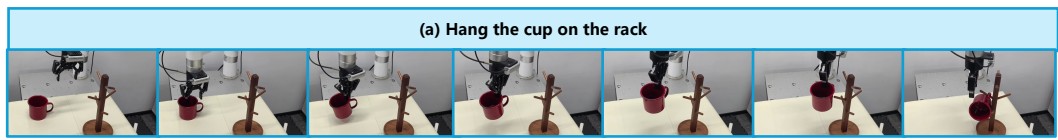

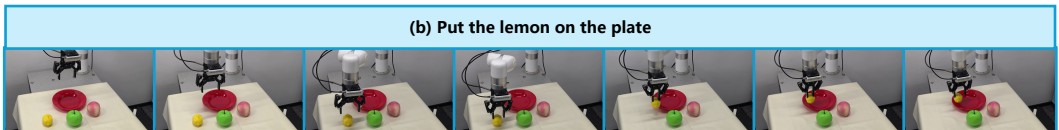

Figure 2: The demonstrations of real-world experiments.

Table 1: The results on $\pi 0$-FAST model.

| Methods | Square |
|---|---|
| Base policy | 85% |
| Dagger | 85% |
| TPO | 90% |
| Ours | **95%** |

Table 2: The results on real-world experiments.

| Methods | Hang | Put |
|---|---|---|
| Base policy | 70% | 85% |
| Dagger | 65% | 85% |
| TPO | 75% | 80% |
| Ours | **90%** | **100%** |

## 4.2 More real-world tasks

In this section, we provide two more real-world experiments as shown in Figure 2. For each task, we collect 100 expert demonstrations to train the base policy. Further, we deploy the base policy to interact with environments and collect 20 human-intervened trajectories. We mix the 20 human-intervened trajectories with 20 expert demonstrations for model preference optimization. As shown in Table 2, our method could achieve better performance compared with other behavior cloning and preference optimization methods.

