# OpenReview forum: "Human-assisted Robotic Policy Refinement via Action Preference Optimization"
_NeurIPS.cc/2025/Conference — NeurIPS 2025 poster_

### Official Review · Reviewer_Qjwp · 2025-06-05

**Clarity:** 3
**Significance:** 2
**Originality:** 2
**Rating:** 4
**Confidence:** 4

**Summary:**

The paper proposes to leverage the intervention of humans during robot manipulation as a preference to guide the learning of the policy for robust behavior.

With a base policy trained on demonstrations, a human will take over when the robot encounters problems, forming an intervention dataset. The base policy will be iteratively fine-tuned on the collected data with preference-based optimization. KL divergence is applied to make sure the policy do not shift too much from the base policy. A reweighting method helps the learning focus on important data points.

Evaluation in both simulation and the real world shows improvement compared to BC methods and other preference-based methods.

**Questions:**

Please see the weaknesses. I am willing to raise the score if my concerns are solved.

**Ethical Concerns:**

["NO or VERY MINOR ethics concerns only"]

**Final Justification:**

During rebuttal. The author clarified the difference with previous work, which addresses my concern about the novelty. The reweighting design also makes more sense to me now.

**Limitations:**

yes.

**Paper Formatting Concerns:**

No.

**Quality:**

3

**Strengths And Weaknesses:**

# Strengths
1. The idea of making VLA continuously improve during deployment while minimizing the human efforts is well-motivated.
2. The setting of the experiments and corresponding analysis is great.
3. The writing is clear and easy to follow.

# Weaknesses
1. My first concern is about the position of this paper. While the authors point out that previous methods learning from human feedbacks [1] necessitate constant human supervision, this paper follows the same way of intervention. Can I understand that the focus of this paper is on perfence-based optimization and reweighting?
2. The design of the reweighting needs further justification. The proposed reweighting is based on the loss term, assuming that more significance should be given to the data that causes bigger shift in the base model. Does this contradict to the KL divergence? How to determine the balance between being adaptive and keeping priors? Besides, it not very clear to me why such reweighting could solve the problem of tokenization. Is there any intuition or mathematical proof behind this?
3. The setting for comparison with baselines is collecting 50 trajectories with intervention and optimize. I am wondering the performance difference if there are more iterations of collection-optimized-collection but keep the total number of trajectories to be the same (i.e. 50).

[1] Liu, Huihan, et al. "Robot learning on the job: Human-in-the-loop autonomy and learning during deployment."

---

> ### Author Rebuttal · Authors · 2025-07-31
>
> Thanks for dedicating time and effort to review our work and providing valuable feedback.
>
> $\color{green}Q1:$ position and focus of this paper.
>
> $\color{blue}A1:$
> Our vision is to enable VLA models to serve the real world, which requires that VLA models can be **deployed reliably** and **continuously improve their capabilities** in real-world scenarios.
>
> However, prior interactive imitation learning works (Sirius) focus on the behavior cloning method, which cannot fully utilize the failure actions of the human-robot interaction trajectory and struggle to retain the prior knowledge for robotic lifelong learning on large-scale VLA models. While existing RL-based methods encounter a challenging optimization landscape in training large-scale VLA models, due to the inherent instability and challenge of developing generalizable value functions.
>
> Thus, we propose to build a framework to ensure **reliable continual policy optimization for large-scale VLA models**. From technical perspective, we propose the human-assisted action preference optimization method to achieve action-level robotic behavior correction with the reweighting strategy on the preference optimization.
>
> Concretely, our human-assisted action preference optimization method include two components:
>
> + The Human-robot Collaboration Deployment phase would deploy the base policy to interact with environment to collect human-intervented interaction trajectory. Depending on the time of human intervention, we would label the action sequence before human intervention as undesirable action and the others as desirable action.
>
> + The Action Preference Optimization phase would optimize the base policy with collected human-intervened trajectories. To make preference learning method work on VLA models optimizaiton, we propose the following techniques:
>     + We introduce the  Kahneman \& Tversky’s prospect theory for preference optimization with binary desirability signals, which release the demand constrain of paired positive-negative actions under the same observational conditions
>     + We propose a reweighting strategy to bridge the gap between token classification and continuous action regression in autoregressive VLA models, which would adaptively improve the weight of error-prone actions.
>
> Through iterative human-robot collaboration deployment and action preference optimization, we could achieve continual improvement from interaction with environments for autoregressive VLA models.
>
> ---
> $\color{green}Q2:$ The performance with multi-turn deployment-optimization loops.
>
> $\color{blue}A2:$ Thanks for this valuable question. We have conducted the multi-round deployment-optimization loops in the Section 4.4 of our paper. In this lifelong learning experiment, we conduct the deployment-optimization loops for 3 rounds with every 20 interaction trajectories.
>
> The results in Coffee_D0 of Figure 3 in our paper are shown below:
> | Base Policy | Round 1 | Round 2 | Round 3 |
> |---|---|---|---|
> | 44 | 54 | 56 | 62 |
>
> The proportions of different action types in each round in Coffee_D0:
> | Action Type  | Round 1 | Round 2 | Round 3 |
> |----|--------------|------- |------- |
> |  Failure Action $\frac{c_0}{c_0 + c_1 + c_2}$| 5.17%         | 5.57% | 3.73% |
> |  Autonomous Action $\frac{c_1}{c_0 + c_1 + c_2}$ | 77.44%     | 79.28% | 87.51% |
> |  Human-intervened Action $\frac{c_2}{c_0 + c_1 + c_2}$ | 17.39%       | 15.15% | 8.76% |
>
> The results of Figure 3 in our paper prove that our method could effectively learn from human-intervened data and achieve self-improvement continually.
>
> To comprehensively evaluate the performance of our method, we also conduct a 2-round deployment-optimization loop experiment with 25 interaction trajectories every round. This experiments ensure that the total number of trajectories is 50, the same as the comparison experiment setting. We also evaluate the performance with 3 random seeds, and report the mean success rate and standard deviation in the table below.
>
> | Base Policy | Round 1 | Round 2 |
> |---|---|---|
> | $44.7 \pm 1.2$ | $54.6 \pm 6.1$ | $60 \pm 4$ |
>
> Although results prove that our method could effectively learn from human-intervened data and achieve self-improvement continually, performance of multi-round deployment-optimization loops is only comparable to the single-round deployment-optimization loop in Table 1 of our paper ($60 \pm 2$).
> A possible explanation is that the failure cases in the target task are highly similar across rounds, so additional deployment-optimization loops do not yield significant improvements over a single round. Thanks for this question to promote us rethink and improve the design of multi-round deployment-optimization loops in further work.
>
> ---
> $\color{green}Q3:$ Explanation of reweighting method. How to determine balance between being adaptive and keeping priors?
>
> $\color{blue}A3:$ Thank you for this valuable question about the proposed adaptive reweighting method.
>
> ### 1. Explaination of reweighting methods and problem of tokenization:
>
> First, we clarify the motivation behind our reweighting method. The core motivation of our action-level preference optimization approach is to encourage the model to learn correct behaviors while avoiding failure actions. Traditional preference optimization methods in LLMs focus only on token probabilities, ignoring the semantic meaning of the decoded words. For instance, if the ground-truth token is _50_, both _10_ and _49_ are treated as equally incorrect.
>
> However, in robot decision making with discrete action space, the continuous action space is encoded to discrete tokens, where each token corresponds to a specific physical action. For example, the action token $i$ represents 'move along the x-axis by $0.i$ cm'. Therefore, it is inappropriate to treat token _10_ and _49_ as equally incorrect. For instance, the desirable action token _50_ may represent "move along the x-axis by 5 cm", while token _49_ corresponds to "move along the x-axis by 4.9 cm", and token _10_ to "move along the x-axis by 1 cm". Clearly, action token _49_ is much closer to the desired action than _10_.
>
> Our reweighting method is designed to leverage this continuous action information for discrete action token preference optimization. This approach bridges the gap between discrete token prediction and continuous action regression, making preference optimization more effective for autoregressive VLA models.
>
> ### 2. The balance between being adaptive and keeping priors:
>
> Thanks for this insightful question, which allows for a deeper exploration of the mechanism behind our reweighting method.
>
> First, it is important to clarify that reweighting method does not contradict the KL divergence constraint. The adaptive coefficients $\lambda_U$ and $\lambda_D$ are sample-wise weights with batch-level normalization, and are estimated using the L1 loss between the ground-truth action $a_{gt}$ and the predicted action $a_{pred} = \pi_{\theta}(o_t)$. While the KL divergence term ensures that the updated policy $\pi_{\theta}$ stays close to the reference policy $\pi_{ref}$. When $\pi_{ref}$ is a good prior, the reweighting method would also encourage the model to preserve prior knowledge.
>
> In fact, the sample-wise reweighting offers an effective trade-off between adaptability and retaining prior knowledge. When the reference model $\pi_{ref}$ provides a good prior, the estimated $l_1$ loss is small, encouraging the model to stay close to the prior distribution. Conversely, if $\pi_{ref}$ is not a suitable prior, the $l_1$ loss becomes larger, prompting the model to adapt more actively to the new distribution.
> In this way, our method adaptively balances preserving prior knowledge and learning from new distributions as needed.
>
> Here, we provide an example below showing how $\lambda$ values vary across samples based on their regression errors.
>
> |Type | L1 loss| $\lambda$|
> |---|---|---|
> | Undesirable Action | [0.05, 0.25] | [0.83, 0.44] |
> | Desirable Action | [0.05, 0.01, 0.39, 0.0, 0.16, 0.15] | [0.06, 0.02, 0.40, 1.2e-4, 0.18, 0.18] |
>
> The values of $\lambda_U$ and $\lambda_D$ indicate that higher weights are assigned to desirable samples with large regression errors and to undesirable samples similar to failure actions.
>
> To assess whether our adaptive reweighting method is effective, we conducted experiments using fixed values: $\lambda_U = 0.5$; $\lambda_D = 1$ for human-intervened actions and 0.5 for autonomous actions to emphasize the human-intervened action. Results are shown below:
>
> Coffee_D0:
> |Method | Success Rate|
> |---|---|
> |HAPO w/o. reweighting | $51.3 \pm 3.0$ |
> |HAPO | $60.0 \pm 2$ |
>
>
> StackThree_D0
> |Method | Success Rate|
> |---|---|
> |HAPO w/o. reweighting | $45.3 \pm 6.1$ |
> |HAPO | $53.3 \pm 5$ |
>
> Square_D0:
> |Method | Success Rate|
> |---|---|
> |HAPO w/o. reweighting | $30.0 \pm 3.5$ |
> |HAPO | $31.3 \pm 3.1$  |
>
> ThreePiece_D0
> |Method | Success Rate|
> |---|---|
> |HAPO w/o. reweighting | $38.7 \pm 2.3$ |
> |HAPO | $43.3 \pm 5.0$ |
>
> These results prove that adaptive, sample-wise reweighting could lead to significantly better performance.
>
> ### 3. Analysis of the reward curves:
> Further, to evaluate whether HAPO effectively promotes learning from human-intervened data via adaptive reweighting, we track the reward of KTO and HAPO during training. The reward is computed as $r_{\theta}(o, a) = \log \frac{\pi_\theta(a|o)}{\pi_{ref}(a|o)}$, where a higher reward for desirable actions indicates stronger model confidence, and a lower reward for undesirable actions suggests better avoidance.
>
> Desirable Reward
>
> | Method| Iter. 1000 | Iter. 2000 | Iter. 3000 |
> |---|---|---|---|
> | KTO | 11.0 | 11.8 | 15.7 |
> | HAPO | 10.2 | 12.9 | 17.4 |
>
> Undesirable Reward
>
> | Method| Iter. 1000 | Iter. 2000 | Iter. 3000 |
> |---|---|---|---|
> | KTO | -10.5 | -18.9 | -26.6 |
> | HAPO | -18.7 | -24.9 | -35.5 |
>
> The reward curves demonstrate that HAPO effectively reweights training samples, encouraging the model to learn more valuable interaction data.

---

> > ### Comment · Reviewer_Qjwp · 2025-08-03
> >
> > Thank the authors for the rebuttal. It addresses my concern about the novelty and reweighting design. I will raise my score to 4.

---

> ### Author Response · Authors · 2025-08-04
> **Thanks for review**
>
> Dear Reviewer,
>
> We sincerely thank you for your time, careful review of our work, and positive feedback on our paper.
>
> Please let us know if you need any further clarification or additional questions.

---

### Official Review · Reviewer_3Xzy · 2025-07-02

**Clarity:** 3
**Significance:** 3
**Originality:** 2
**Rating:** 5
**Confidence:** 3

**Summary:**

The authors introduce HAPO, a preference-tuning algorithm that enables interactive imitation learning for VLAs. Whilst the goal and some of the pipeline is similar to prior algorithms, such as Sirius, the authors show that adapting VLAs require more specialised update rules. The authors show the benefits of their algorithm against baselines, generalisation and continual learning, application to more than one VLA, and real-world robot experiments.

**Questions:**

How were hyperparameter chosen for the different methods in the experiments? It's not clear that every method would perform best with the same set of hyperparameters.

**Ethical Concerns:**

["NO or VERY MINOR ethics concerns only"]

**Final Justification:**

I have read the other reviews and the authors' responses, and believe that my rating of 5 is warranted due to the contributions and depth of the work.

**Limitations:**

Yes.

**Paper Formatting Concerns:**

None.

**Quality:**

2

**Strengths And Weaknesses:**

Given some experience with robot learning, I feel the main strength of this work is the experimental section, which demonstrates the performance improvement given by using their algorithm, as opposed to various more model-agnostic algorithms and other preference tuning algorithms.

Initially, I would have questioned the advancement of HAPO over Sirius, but the authors do use Sirius as a baseline and show that their method is better for optimising VLAs. Nowadays most robot learning policies tend to use diffusion over autoregressive prediction, but I would consider this beyond the scope of this work (the authors also acknowledge this in the limitations section).

The authors could expand the explanation of HAPO to make it easier to grasp, but given space limitations the current size of the various sections are reasonable.

---

> ### Author Rebuttal · Authors · 2025-07-31
>
> Thank you for recognizing our work. We sincerely appreciate the time and effort you dedicated to reviewing our paper and providing valuable feedback.
>
> $\color{green}Q1:$ The advancement of HAPO over Sirius.
>
> $\color{blue}A1:$
> Thanks for this question. We believe Sirius is a great and solid work in the area of human-robot collaboration and continual policy learning.
>
> The award of this paper (RSS 2023 Best Paper Finalist) demonstrates the embodied AI community's strong emphasis on the **reliable robotic deployment and lifelong learning of robot policies** in real-world scenarios.
>
> However, Sirius relies on behavior cloning to continually optimize its policy, which leads to several limitations listed below. Thus, we propose a human-assisted action preference optimization method to solve these issues:
>
> 1. Behavior cloning cannot effectively leverage failed actions from human-robot interaction trajectories, even though these are highly informative for teaching the model what to avoid. Our preference optimization method **incorporates these failure actions to guide learning away from undesired behaviors**.
>
> 2. As shown in the experiment results, the behavior cloning method is struggling to both retain the prior knowledge and adapt to new distribution shifts of interaction trajectories. While our method utilize the reference model to constrain the policy optimization for prior knowledge retain, and utilize a reweighting strategy to **encourage model to avoid failure actions and correct from failure cases**.
>
> The experiment results demonstrate that our mehtod could achieve better performance on various novel downstream tasks compared with Sirius and other baselines, different autoregressive VLA models and real-world tasks.
>
> ---
> $\color{green}Q2:$ The difference between diffusion policy and autoregressive policy.
>
> $\color{blue}A2:$
> Thanks for this question. It is a hot and important topic about the optimal architecture for VLA models.
> Due to the lack of robotic data and the wonderful generalizability of VLMs, no matter the diffusion-based or autoregressive-based VLA models are mainly built on the VLM backbone. For example, the OpenVLA[1], RT-2[2], $\pi_0$-FAST[3] encode the action to discrete tokens and model the action prediction task as an autoregressive task. While $\pi_0$[4], Go-1[5] utilize flow-based models as an additional action expert for robotic action prediction.
>
> However, there is still no definitive conclusion that whether diffusion-based methods or autoregressive methods are better for generating robotic actions. Although the diffusion-based methods have better ability to build the multi-modal action distribution, recent papers show that the autoregressive methods can also achieve good performance in robotic action prediction.
> For example, the OpenVLA-OFT[6] show that the OpenVLA model can also achieve better performance than $\pi_0$ with the action chunking trick. While the knowledge insulation[7] paper shows that the gradient of the flow-based model would harm the knowledge retention of the VLM.
>
> In our work, the proposed HAPO method is mainly designed for the autoregressive model, and we are also interested in how to apply our method to the diffusion-based model, which would be a future work. Thanks for this valuable question.
>
>
>
> ---
> $\color{green}Q3:$ The motivation and explanation of our human-assisted action preference optimization.
>
> $\color{blue}A3:$
> Thanks for this question. We would provide a more comprehensive description of the motivation and explanation in the revised version of this work.
>
> Our vision is to enable robots to truly serve the real world, which requires that **robots can be deployed reliably and continuously improve their capabilities** in real-world scenarios.
>
> Prior works (Sirius) on human-in-the-loop learning mainly focus on the behavior cloning method, which cannot fully utilize the failure actions of the human-robot interaction trajectory and struggle to retain the prior knowledge for robotic lifelong learning on large-scale VLA models.
>
> Thus, we propose the human-assisted action preference optimization method to correct action-level robotic behavior to solve these issues as mentioned in the Answer 1.
>
> Concretely, our HAPO method include two components:
>
> + The **Human-robot Collaboration Deployment phase** would deploy the base policy to interact with environment to collect human-intervented interaction trajectory. Depending on the time of human intervention, we would label the action sequence before human intervention as undesirable action and the others as desirable action.
>
> + The **Action Preference Optimization phase** would optimize the base policy with collected human-intervened trajectories. To make preference learning method work on VLA models optimizaiton, we propose the following techniques:
>     + We introduce the  Kahneman \& Tversky’s prospect theory for preference optimization with binary desirability signals, which release the demand constrain of paired positive-negative actions under the same observational conditions
>     + We propose a reweighting strategy to bridge the gap between token classification and continuous action regression in autoregressive VLA models, which would adaptively improve the weight of error-prone actions.
>
> Through iterative human-robot collaboration deployment and action preference optimization, we could achieve continual improvement from interaction with environments for autoregressive VLA models.
>
> ---
> $\color{green}Q4:$ The hyperparameter for different models.
>
> $\color{blue}A4:$ Thanks for this suggestion, we would provide a more comprehensive description of the implementation details in the further vision of this work.
>
> For all baselines, we first deploy the base model $\pi_\theta$ on the environment to collect 50 human-intervened interaction trajectories. Subsequently, we mix these 50 human-intervened trajectories with 50 expert demonstrations to fine-tune the base model using different baseline methods. All methods are trained with a learning rate of 5e-5. For BC-based methods, we use the observation-action pairs (o_t, a_t) with $c_t \neq 0$ as supervised learning samples. While for preference learning based methods, we recognize the observation-action pairs (o_t, a_t) with $c_t = 0$ as undesirable samples and recognize the observation-action pairs (o_t, a_t) with $c_t \neq 0$ as desirable samples.
>
> Besides, it is an interesting question to explore the influence of failure action selection on the performance of different methods, because the selection of $K$ plays a crucial role in identifying which actions should be considered failures for the model to avoid. We provide the performance of different methods with different $K$ values in the following table.
>
> Results on the Coffee_D0 task:
>
> | Task Name   | $K=5$  | $K=10$ | $K=15$ | $K=20$ |
> |-------------|------|------|------|------|
> | KTO   | $48.6 \pm 4.2$ | $52 \pm 3.4$ | $54 \pm 6$  | $54 \pm 11.1$ |
> | HAPO (ours)   | $54.6 \pm 6.1$ | $60.0 \pm 2$ | $59.3 \pm 7.5$  | $63.3 \pm 4.6$ |
>
>
> Results on the StackThree_D0 task:
>
> | Task Name   | $K=5$  | $K=10$ | $K=15$ | $K=20$ |
> |-------------|------|------|------|------|
> | KTO   | $51.3 \pm 4.6$ | $50 \pm 5.3$ | $50.6 \pm 4.6$  | $50.6 \pm 4.1$ |
> | HAPO (ours)    | $51.3 \pm 2.3$ | $53.3 \pm 5$| $56.6 \pm 4.6$ | $54.6 \pm 5.0$  |
>
> The results prove that our method could achieve consistent performance improvement than other baselines on different tasks.
>
>
> ---
> [1] OpenVLA: An Open-Source Vision-Language-Action Model.
>
> [2] RT-2: Vision-Language-Action Models Transfer Web Knowledge to Robotic Control.
>
> [3] FAST: Efficient Robot Action Tokenization.
>
> [4] $\pi_0$: Our First Generalist Policy.
>
> [5] Agibot world colosseo: A large-scale manipulation platform for scalable and intelligent embodied systems.
>
> [6] Fine-Tuning Vision-Language-Action Models: Optimizing Speed and Success.
>
> [7] Knowledge Insulating Vision-Language-Action Models: Train Fast, Run Fast, Generalize Better.

---

> ### Author Response · Authors · 2025-08-04
> **Thanks for review**
>
> Dear Reviewer,
>
> We are deeply grateful for your thoughtful review and the generous time you have devoted to our paper. Your constructive feedback is immensely valuable to us.
>
>  If you would like any clarification or have further questions, we would be delighted to provide additional information.

---

> > ### Comment · Reviewer_3Xzy · 2025-08-05
> > **Rebuttal Response**
> >
> > Thank you for the rebuttal. I have read the other reviews and responses and will be keeping my score at 5.

---

> > > ### Author Response · Authors · 2025-08-06
> > > **Thanks for review**
> > >
> > > Dear reviewer,
> > >
> > > We are profoundly grateful for your constructive review of our work and your appreciation of our work.

---

### Official Review · Reviewer_ScuL · 2025-07-02

**Clarity:** 2
**Significance:** 3
**Originality:** 3
**Rating:** 4
**Confidence:** 3

**Summary:**

This paper proposes a VLA optimization method, Human-assisted Action Preference Optimization (HAPO), which includes a human-robot collaboration data collection framework and an adaptive reweighting algorithm. The authors claim that the proposed adaptive reweighting algorithm addresses intrinsic issues in VLA optimization—specifically, the irreversible nature of interactions during data collection and the mismatch between token probabilities and continuous action errors. The paper presents extensive experiments demonstrating that HAPA outperforms DPO, KPO, TPO, etc., in optimizing VLAs, improving robustness to dynamic environmental changes, and generalizing across different VLA architectures.

**Questions:**

1. **Trade-off for more precise and fine-grained optimization**: Since HAPO uses Prospect Theory to construct binary labels instead of pairwise data for policy optimization, one concern is that providing only coarse supervision—i.e., judging whether a trajectory is good or bad—may be too vague. In this case, the model might learn undesirable shortcuts or stereotyped behaviors that minimize the loss, rather than learning diverse and robust motion patterns to achieve the goal.
2. **Questionable quantitative results**: Based on Table 1, the overall performance gain of HAPO over KTO does not appear to be very substantial. Without error bars, it is difficult to assess whether the reported improvements are statistically meaningful. Furthermore, it seems that most of the gains come primarily from the Coffee_D0 task, which raises concerns about statistical significance and broader applicability.

**Ethical Concerns:**

["NO or VERY MINOR ethics concerns only"]

**Final Justification:**

The work conducts a wide range of experiments and ablations to evaluate HAPO—not only for its core function in model optimization, but also for its robustness to dynamic environmental changes and its generalization across different VLAs. I believe all of these are important metrics for assessing a VLA optimization method. During the rebuttal, the authors have conducted extensive additional experiments to address my questions, including the statistical significance and additional explanation of reweighting methods. Therefore, I recommend acceptance of this paper.

**Limitations:**

Please refer to the Weaknesses section to address the primary concerns. I also have the following additional questions:

- As mentioned above, Prospect Theory introduces concerns regarding the use of coarse supervision signals. I’m curious that when humans collaborate with the robot to create new trajectories based on the failed ones from the base model, what kind of behavior do they follow? Do they try to mimic the failed trajectory but correct it slightly, or do they focus entirely on achieving the goal, without caring about replicating the original path?

- The paper also claims to address the token mismatch issue through adaptive reweighting (Equations 4–6), but lacks an ablation study isolating the benefit of this specific component. It would be helpful to understand how much performance gain can be attributed directly to this reweighting mechanism.

- Finally, regarding statistical significance: across both the simulation and real-world experiments, the paper presents results on a limited set of tasks and without error bars. This is particularly concerning given that the performance boost of HAPO over the baselines is not very large in absolute terms. More rigorous statistical reporting would help support the claimed improvements.

**Quality:**

2

**Strengths And Weaknesses:**

The work conducts a wide range of experiments and ablations to evaluate HAPO—not only for its core function in model optimization, but also for its robustness to dynamic environmental changes and its generalization across different VLAs. I believe all of these are important metrics for assessing a VLA optimization method.

---

> ### Author Rebuttal · Authors · 2025-07-31
>
> Thank you for recognizing our work. We sincerely appreciate your time and effort in reviewing our work and offering insightful feedback.
>
> $\color{green}Q1:$ Trade-off for more precise and fine-grained optimization.
>
> $\color{blue}A1:$
> Thanks for this valuable question. One of the motivations of our work is to **provide fine-grained action correction for the VLA models**. To encourage the model to learn how to avoid the failure action and achieve reliable failure correction, we design the human-assisted action preference optimization (HAPO) method, which learns from the **fine-grained action-level binary desirability signals** instead of the trajectory-level coarsed-grained signals.
>
> Specifically, as illustrated in Algorithm 1, during the deployment phase, we first deploy the base model $\pi_\theta$ in the environment to collect human-intervened interaction trajectories. These trajectories are then re-labeled with binary action-level desirability signals based on the timing of human interventions: the $K$ steps preceding each intervention are considered undesirable action sequences.
>
> To ensure that the preference optimization method is well-suited for VLA models, we incorporate Kahneman \& Tversky’s prospect theory to guide the optimization process with fine-grained binary desirability signals. We also propose a reweighting strategy to bridge the gap between discrete token prediction and continuous action regression, promoting the model to pay more attention to the fine-grained error-prone actions.
>
> Through the above algorithm, our human-assisted action preference optimization method can **learn from the action-level binary desirability signals to provide fine-grained action correction for the VLA models**. We also provide the supplementary videos to show the learned correction patterns of our model.
>
> ---
> $\color{green}Q2:$ The behavior of the our human-assisted action preference optimization method.
>
> $\color{blue}A2:$
> It is an interesting question to check the behavior of our human-assisted action preference optimization method. We observe that the robot often fails on some similar failure modes. For instance, in the "Coffee_D0" simulation task, the most common failure involves the model being unable to insert the coffee pod into the machine. Similarly, in the "Insert" real-world task, the model often fails to align the stick with the square hole.
>
> During the deployment phase, we manually intervene to correct such failures. These failure actions are labeled as undesirable data, while the corresponding human-intervened actions are treated as desirable data for preference optimization.
>
> By learning corrective actions from human interventions, the model effectively adapts to downstream tasks and avoids these failure patterns, even with limited data.
> As demonstrated in the supplementary videos, the model optimized via our HAPO method is capable of recovering from these failure scenarios. For example, in the Coffee_D0 task, the model learns to slightly adjust the gripper position to successfully insert the coffee pod. In the real-world square insertion task, the model learns to retry the alignment when the stick is initially misaligned with the hole. For a more comprehensive understanding of our method’s behavior, please refer to the supplementary videos.
>
> ---
> $\color{green}Q3:$ More quantitative results with statistical significance.
>
> $\color{blue}A3:$
> Thanks for this valuable question. In this work, to comprehensively evaluate the performance of our method, we compared our HAPO methods with other baselines in two settings: the in-distribution setting, the out-of-distribution setting.
>
> In the in-distribution setting, to thoroughly validate the effectiveness of our method, we report the average success rate and standard deviation for simulation tasks in Table 1, based on 50 rollouts with 3 different seeds. Additionally, we introduce a new simulation task, "Cleanup Mug," and a real-world task, "Stack Blocks," to further assess our method's performance. The "Cleanup Mug" task is a long-horizon task in which the robot must open a drawer, place a mug inside, and then close the drawer. The "Stack Blocks" task is a fine-grained task that requires the robot to stack a 2cm block on top of another. For real-world tasks, we perform 20 rollouts. The results are shown in the following table:
>
> Coffee_D0:
> |Method | Success Rate|
> |---|---|
> |Base Model | $44.7 \pm 1.2$ |
> |KTO | $48.6 \pm 4.2$ |
> |HAPO | $60.0 \pm 2$ |
>
>
> StackThree_D0:
> |Method | Success Rate|
> |---|---|
> |Base Model | $42 \pm 7.2$ |
> |KTO | $50.7 \pm 4.2$ |
> |HAPO | $53.3 \pm 5$ |
>
> Square_D0:
> |Method | Success Rate|
> |---|---|
> |Base Model | $28.0 \pm 4.0$ |
> |KTO | $28.6 \pm 3.1$ |
> |HAPO | $31.3 \pm 3.1$  |
>
> ThreePiece_D0:
> |Method | Success Rate|
> |---|---|
> |Base Model | $ 39.3 \pm 5.0$ |
> |KTO | $41.3 \pm 5.7$ |
> |HAPO | $43.3 \pm 5.0$ |
>
> Real World Stack Block(New):
> |Method | Success Rate|
> |---|---|
> |Base Model | $ 48.3 \pm 2.9$ |
> |KTO | $51.7 \pm 2.9$ |
> |HAPO | $63.3 \pm 5.7$ |
>
> Simulation Cleanup Mug(New):
> |Method | Success Rate|
> |---|---|
> |Base Model | $ 30.6 \pm 9.8$ |
> |KTO | $36.6 \pm 3.0$ |
> |HAPO | $42 \pm 8.7$ |
>
>
> In the out-of-distribution setting, the results in Table 2 and Table 3 of our paper show that our method achieves consistent improvement over the baseline methods in most scenarios. We also evaluate the baseline methods for 50 rollouts with 3 different seeds and report the average success rate and standard deviation to validate the stability of our method:
>
> Background Disruption:
> |Method | Success Rate|
> |---|---|
> |Base Model | $39.3 \pm 4.2$ |
> |KTO | $40.6 \pm 3.0$ |
> |HAPO | $44.7 \pm 1.1$ |
>
> The results show that our method could achieve consistent improvement over the baseline methods in both in-distribution and out-of-distribution settings with statistical significance.
>
> ---
> $\color{green}Q4:$ The analysis of the reweighting method.
>
> $\color{blue}A4:$
> Thank you for this valuable question about the proposed adaptive reweighting method.
>
> ### 1. Explaination of the reweighting methods:
> First, we would like to clarify the motivation behind our reweighting method. The core motivation of our action-level preference optimization approach is to encourage the model to learn correct behaviors while avoiding failure actions. Traditional preference optimization methods in LLMs focus solely on token probabilities and overlook the semantic meaning of the decoded words. For example, if the ground-truth token is _50_, both _10_ and _49_ would be considered equally incorrect.
>
> However, in robot decision making with discrete action space, the continuous action space is encoded to discrete tokens, which means different tokens correspond to different physical actions. For example, the action token $i$ represents 'move along the x-axis by $0.i$ cm'. Therefore, it is inappropriate to treat token _10_ and _49_ as equally incorrect. For instance, the desirable action token _50_ may represent "move along the x-axis by 5 cm", while token _49_ corresponds to "move along the x-axis by 4.9 cm", and token _10_ to "move along the x-axis by 1 cm". Clearly, action token _49_ is much closer to the desired action than _10_.
>
> Our reweighting method is designed to leverage this continuous action information for discrete action token preference optimization. This approach bridges the gap between discrete token prediction and continuous action regression, making preference optimization more effective for autoregressive VLA models.
>
> ### 2. Analysis of the reweighting method:
>
> Our $\lambda_U$ and $\lambda_D$ are sample-wise weights with batch-level normalization, designed to prioritize samples with large regression errors. Here, we provide an example below showing how $\lambda$ values vary across samples based on their regression errors.
>
> |Type | L1 loss| $\lambda$|
> |---|---|---|
> | Undesirable Action | [0.05, 0.25] | [0.83, 0.44] |
> | Desirable Action | [0.05, 0.01, 0.39, 0.0, 0.16, 0.15] | [0.06, 0.02, 0.40, 1.2e-4, 0.18, 0.18] |
>
> The values of $\lambda_U$ and $\lambda_D$ indicate that higher weights are assigned to desirable samples with large regression errors and to undesirable samples similar to failure actions.
>
> To assess whether our adaptive reweighting method is effective, we conducted experiments using fixed values: $\lambda_U = 0.5$; $\lambda_D = 1$ for human-intervened actions and 0.5 for autonomous actions to emphasize the human-intervened action. Results are shown below:
>
> Coffee_D0:
> |Method | Success Rate|
> |---|---|
> |HAPO w/o. reweighting | $51.3 \pm 3.0$ |
> |HAPO | $60.0 \pm 2$ |
>
> StackThree_D0:
> |Method | Success Rate|
> |---|---|
> |HAPO w/o. reweighting  | $45.3 \pm 6.1$ |
> |HAPO | $53.3 \pm 5$ |
>
> Square_D0:
> |Method | Success Rate|
> |---|---|
> |HAPO w/o. reweighting | $30.0 \pm 3.5$ |
> |HAPO | $31.3 \pm 3.1$  |
>
> ThreePiece_D0
> |Method | Success Rate|
> |---|---|
> |HAPO w/o. reweighting | $38.7 \pm 2.3$ |
> |HAPO | $43.3 \pm 5.0$ |
>
> These results prove that adaptive, sample-wise reweighting could lead to significantly better performance.
>
> ### 3. Analysis of the reward curves:
> Further, to evaluate whether HAPO effectively promotes learning from human-intervened data via adaptive reweighting, we track the reward of KTO and HAPO during training. The reward is computed as $r_{\theta}(o, a) = \log \frac{\pi_\theta(a|o)}{\pi_{ref}(a|o)}$, where a higher reward for desirable actions indicates stronger model confidence, and a lower reward for undesirable actions suggests better avoidance.
>
> Desirable Reward
>
> | Method| Iter. 1000 | Iter. 2000 | Iter. 3000 |
> |---|---|---|---|
> | KTO | 11.0 | 11.8 | 15.7 |
> | HAPO | 10.2 | 12.9 | 17.4 |
>
> Undesirable Reward
>
> | Method| Iter. 1000 | Iter. 2000 | Iter. 3000 |
> |---|---|---|---|
> | KTO | -10.5 | -18.9 | -26.6 |
> | HAPO | -18.7 | -24.9 | -35.5 |
>
> The reward curves demonstrate that HAPO effectively reweights training samples, encouraging the model to learn more valuable interaction data.

---

> ### Author Response · Authors · 2025-08-04
> **Thanks for review**
>
> Dear Reviewer,
>
> We sincerely thank you for your time and careful review of our work.
>
> We have thoroughly addressed questions and suggestions raised during the review process. Below we provide detailed point-by-point responses to specific questions:
>
> 1. Trade-off for more precise and fine-grained optimization.
>
> 2. The behavior of our human-assisted action preference optimization method.
>
> 3. More quantitative results with statistical significance.
>
> 4. The analysis of the reweighting method.
>
> Please let us know if you need any further clarification or additional questions.

---

> ### Comment · Area_Chair_U3ZE · 2025-08-09
> **Comment by AC**
>
> Dear Reviewer,
>
> Thank you for your participation in the review process. If you haven't done these steps, please engage in the discussion phase by following these guidelines:
>
> - Read the author rebuttal;
> - Engage in discussions;
> - Fill out the "Final Justification" text box and update the "Rating" accordingly.
>
> Reviewers must participate in discussions with authors before submitting “Mandatory Acknowledgement”. The deadline is Aug 8, 11.59pm AoE.
>
> Thanks,
>
> AC

---

### Official Review · Reviewer_uJj6 · 2025-07-05

**Clarity:** 3
**Significance:** 2
**Originality:** 2
**Rating:** 4
**Confidence:** 3

**Summary:**

This paper proposes Human-assisted Action Preference Optimization (HAPO) method to improve the deployment reliability and adaptation of Vision-Language-Action (VLA) models in robotic manipulation tasks. HAPO uses a weighted action preference optimization method and an reweighting method to bridge the gap between discrete token prediction and continuous action regression. Experiments in both simulation and real-world demonstrate its effectiveness.

**Questions:**

- In the real-world experiments, what is the total number of human-labeled trajectories needed for significant improvements?

- Since the binary labeling (desirable vs undesirable) is based on K-step pre-intervention windows, could the authors comment on how sensitive is performence to the choice of K?

**Ethical Concerns:**

["NO or VERY MINOR ethics concerns only"]

**Final Justification:**

The authors address my initial concerns:

1. The missing loss functions of the baseline and re-implementation details of DAgger.

2. More analysis on the intrinsic mechanism of HAPO, including the proportions of different action types, ablation study of the reweighting method, and the reward curves of HAPO. These give me a more comprehensive understanding of the underlying mechanism of the proposed method, which was the most important reason for my score increase.

**Limitations:**

See weaknesses and questions.

**Paper Formatting Concerns:**

No major formatting issues

**Quality:**

3

**Strengths And Weaknesses:**

Strengths:
- HAPO helps to bridge the gap between discrete token prediction and continuous action regression.

- The authors conducted throughout experiments both in simulation and real-world, their conclusions are well supported by convincing experimental results.

- The paper is generally well-organized.

Weaknesses:
- Experimental details of the baseline are missing. Suggest adding details about the baseline, such as specific loss function formulas.

- There may be non-rigorous baseline re-implementation. If Dagger is re-implemented properly, then it is unlikely to be worse than the base model. There may be non-rigorous experimental details such as: did you use data with $c_t \neq 0$ for dagger? Since $c_t=0$ is known to be undesirable data, using such data would be a targeted poisoning of the behavioral cloning methods.

- Many experiments to show that HAPO is better than the baseline in various scenarios (e.g., simulation, real-world, novel tasks, lifelong learning, other VLA, etc.) is nice, of course, but discussions of the intrinsic mechanism of HAPO is missing. For example:

    - Let me illustrate with pseudo-code (Algorithm 1), what are the proportions of $c_t=0,1,2$ in each loop (line 9) respectively? Does the proportion of human intervention gradually decreases as training proceeds? Please draw a figure with iteration $i$ as the horizontal axis and proportion as the vertical axis, with three curves, including $c_0/(c_0+c_1+c_2)$, $c_1/(c_0+c_1+c_2)$, and $c_2/(c_0+c_1+c_2)$.

    - What are the curves of $\lambda_D$ and $\lambda_U$? Do they both vary much during training? If not much, is it possible that I can solve the reweighting issues by setting a good and fixed hyperparameter value for $\lambda_D$ and $\lambda_U$?

If the author could provide more analysis and experiments on the mechanisms inherent in HAPO, I would be willing to raise the score.

---

> ### Author Rebuttal · Authors · 2025-07-31
>
> Thanks for dedicating time and effort to review our work and providing valuable feedback.
>
> $\color{green}Q1:$ The experimental details of the baseline.
>
> $\color{blue}A1:$
> Thanks for this suggestion, we will provide a more comprehensive description in the revised version.
>
> For all baselines, we deploy base model $\pi_\theta$ to collect 50 human-intervened trajectories and mix them with 50 expert demonstrations for fine-tuning. All methods use a learning rate of 5e-5. For BC-based methods, we treat observation-action pairs $(o_t, a_t)$ with $c_t \neq 0$ as supervised learning samples. For preference-based methods, pairs with $c_t \neq 0$ are desirable, and those with $c_t = 0$ are undesirable.
>
> Below are loss functions and implementation details. For brevity, we only show desirable sample loss for KTO and HAPO:
>
> | Method | Loss Function | Implementation Details |
> |--------|---------------|------------------------|
> | Dagger | $L_{dagger}(\pi_\theta) = -\sum_{(o_i,a_i)\in D} \log \pi_\theta(a_i\|o_i)$  |  |
> | Sirius |  $ L_{sirius}(\pi_\theta) = -\sum_{(o_i,a_i)\in D} w_i \log \pi_\theta(a_i\|o_i)$ | $w_i$ is estimated by ratio of number of human-intervened actions.|
> | DPO | $L_{DPO}(\pi_\theta; \pi_{ref}) = -E_{(o_i,a_d,a_u) \sim D}[\log \sigma( \log(\frac{\pi_\theta(a_d\|o_i)}{\pi_{ref}(a_d\|o_i)} - \log(\frac{\pi_\theta(a_u\|o_i)}{\pi_{ref}(a_u\|o_i)}))]$ | undesirable action $a_u$ is generated by base model $\pi_{ref}$ on observation $o_i$ with random action noise. |
> | TPO | $L_{TPO}(\pi_\theta; \pi_{ref}) = -E_{(\zeta_d, \zeta_u) \sim D}[\log \sigma( \log(\frac{\pi_\theta(\zeta_d)}{\pi_{ref}(\zeta_d)} - \log(\frac{\pi_\theta(\zeta_u)}{\pi_{ref}(\zeta_u)}))]$ |  We recognize pre-intervention sub-trajectory as undesirable sub-trajectory and other sub-trajectory as desirable one. |
> | KTO | $L_{KTO}(\pi_\theta; \pi_{ref}) = -E_{(o_i,a_i) \sim D}[ \sigma( \frac{\pi_\theta(a_i\|o_i)}{\pi_{ref}(a_i\|o_i)} - KL(\pi_\theta, \pi_{ref}))]$ |  |
> | HAPO(ours) | $L_{HAPO}(\pi_\theta; \pi_{ref}) = -E_{(o_i,a_i) \sim D}[ (1 - e^{-\beta_D*w_i}) * \sigma( \frac{\pi_\theta(a_i\|o_i)}{\pi_{ref}(a_i\|o_i)} - KL(\pi_\theta, \pi_{ref}))]$| We achieve sample-wise reweighting control on $w_i$ to prioritize error-prone samples. |
>
> The experiments on various downstream tasks validate the effectiveness of our HAPO method, and we also provide more comprehensive analysis in Q3.
>
>
> ---
> $\color{green}Q2:$ The re-implementation of DAgger.
>
> $\color{blue}A2:$ Thanks for this question to ensure fair comparison for all baselines.
>
> We followed Sirius[1] setup to re-implement both DAgger and Sirius, using only observation-action pairs $(o_i, a_i)$ with $c_i \neq 0$ as supervised learning samples in our paper. This ensures a rigorous re-implementation of DAgger.
>
> It is common that the updated model underperforms the base model in interactive imitation learning, as shown in Table 2 of the OLAF paper[2] and the Sirius-Runtime-Monitor paper[3], where HG-DAgger and ThriftyDAgger are variants of DAgger. As discussed in Section 4.1 of our paper, one possible reason for this phenomenon is that BC-based methods often struggle to retain the knowledge of the base model when learning from the distribution-shifted human-intervened data.
>
> The Table 2 results in OLAF paper:
>
> | Method | Success Rate |
> |---|---|
> | Base Model |  84.4 |
> | HG-DAgger |  75.0 |
>
> The Table 2 results in sirius-runtime-monitor paper:
>
> | Method | Round 1 | Round 2 | Round 3 |
> |---|---|---|---|
> | ThriftyDAgger |  79.5 | 75.6 | 77.5 |
>
> ---
> $\color{green}Q3:$ More analysis on intrinsic mechanism of HAPO.
>
> $\color{green}Q3.1:$ Proportions of different action types.
>
> $\color{blue}A3.1:$
> We appreciate this insightful question. We have reported the human-intervention proportions ratio and succcess rate during multi-round deployment-optimization loops in Figure 3 of our paper, demonstrating that our HAPO method continuously improves through interaction with environment. Below is a more detailed table:
>
> Proportions in Coffee_D0:
>
> | Action Type  | Round 1 | Round 2 | Round 3 |
> |----|--------------|------- |------- |
> |  Failure Action $\frac{c_0}{c_0 + c_1 + c_2}$| 5.17%         | 5.57% | 3.73% |
> |  Autonomous Action $\frac{c_1}{c_0 + c_1 + c_2}$ | 77.44%     | 79.28% | 87.51% |
> |  Human-intervened Action $\frac{c_2}{c_0 + c_1 + c_2}$ | 17.39%       | 15.15% | 8.76% |
>
> Proportions in StackThree_D0:
>
> | Action Type  | Round 1 | Round 2 | Round 3 |
> |----|--------------|------- |------- |
> |  Failure Action $\frac{c_0}{c_0 + c_1 + c_2}$| 4.59%         | 4.07% | 3.52% |
> |  Autonomous Action $\frac{c_1}{c_0 + c_1 + c_2}$ | 81.97%     | 83.26% | 85.09% |
> |  Human-intervened Action $\frac{c_2}{c_0 + c_1 + c_2}$ | 13.43%       | 12.67% | 11.39% |
>
> The results prove that our HAPO method could effectively learn from interaction data and achieve self-improvement continually.
>
> $\color{green}Q3.2:$ Curves of $\lambda_U$ and $\lambda_D$. Is it possible to solve the reweighting issues with a good and fixed hyperparameter?
>
> $\color{blue}A3.2:$
> Our $\lambda_U$ and $\lambda_D$ are sample-wise weights with batch-level normalization, designed to prioritize samples with large regression errors. Due to their sample-wise nature, it's difficult to capture trends with a curve. Instead, we provide an example below showing how $\lambda$ values vary across samples based on their regression errors.
>
> |Type | L1 loss| $\lambda$|
> |---|---|---|
> | Undesirable Action | [0.05, 0.25] | [0.83, 0.44] |
> | Desirable Action | [0.05, 0.01, 0.39, 0.0, 0.16, 0.15] | [0.06, 0.02, 0.40, 1.2e-4, 0.18, 0.18] |
>
> The $\lambda_U$ and $\lambda_D$ values show that higher weights are given to desirable samples with large regression errors and undesirable samples similar to failure actions.
>
> To assess whether fixed weights can replace adaptive reweighting, we conducted experiments using fixed values: $\lambda_U = 0.5$; $\lambda_D = 1$ for human-intervened actions and 0.5 for autonomous actions to emphasize human-intervened action:
>
> Coffee_D0:
> |Method | Success Rate|
> |---|---|
> |HAPO w/. fixed weight | $51.3 \pm 3$ |
> |HAPO | $60.0 \pm 2$ |
>
>
> StackThree_D0
> |Method | Success Rate|
> |---|---|
> |HAPO w/. fixed weight | $45.3 \pm 6.1$ |
> |HAPO | $53.3 \pm 5$ |
>
> Square_D0:
> |Method | Success Rate|
> |---|---|
> |HAPO w/. fixed weight | $30.0 \pm 3.5$ |
> |HAPO | $31.3 \pm 3.1$  |
>
> ThreePiece_D0
> |Method | Success Rate|
> |---|---|
> |HAPO w/. fixed weight | $38.7 \pm 2.3$ |
> |HAPO | $43.3 \pm 5$ |
>
> Results prove that our adaptive, sample-wise reweighting is more effective than fixed parameters, leading to significantly better performance.
>
> $\color{green}Q3.3:$ More analysis on the HAPO methods.
>
> $\color{blue}A3.3:$
> To evaluate how HAPO effectively promotes learning from human-intervened data via adaptive reweighting, we track the reward of KTO and HAPO during training. The reward is computed as $r_{\theta}(o, a) = \log \frac{\pi_\theta(a|o)}{\pi_{ref}(a|o)}$, where a higher reward for desirable actions indicates stronger model confidence, and a lower reward for undesirable actions suggests better avoidance.
>
> Desirable Reward
>
> | Method| Iter. 1000 | Iter. 2000 | Iter. 3000 |
> |---|---|---|---|
> | KTO | 11.0 | 11.8 | 15.7 |
> | HAPO | 10.2 | 12.9 | 17.4 |
>
> Undesirable Reward
>
> | Method| Iter. 1000 | Iter. 2000 | Iter. 3000 |
> |---|---|---|---|
> | KTO | -10.5 | -18.9 | -26.6 |
> | HAPO | -18.7 | -24.9 | -35.5 |
>
> The reward curves demonstrate that HAPO effectively reweights training samples, encouraging the model to learn more valuable interaction data.
>
> ---
> $\color{green}Q4:$ The total number of human-labeled trajectories needed for significant improvements in the real-world experiments.
>
> $\color{blue}A4:$ As detailed in Section 4.6 of our paper, we collect 100 expert demonstrations to train the base model, followed by 20 human-intervened trajectories for fine-tuning, leading to significant improvements in downstream tasks.
>
> We observe that robot often fails due to similar failure modes, such as misalignment in the "Insert" task or failed grasps in the "Pick" task, as shown in the supplementary videos. By learning corrective actions from human interventions, the model adapts to these tasks and avoids such failures, even with limited data.
>
> Results in Table 5 of our paper and Table 2 in the supplementary material confirm that our approach enhances both performance and generalization to new disruptions. The supplementary videos further demonstrate the learned correction behaviors.
>
> ---
> $\color{green}Q5:$ How does the pre-intervention window size $K$ affect model performance?
>
> $\color{blue}A5:$
> Thank you for this insightful question. The choice of $K$ determines "failure action" in preference optimization method, plays a crucial role in identifying which actions should be considered failures for model to avoid:
>
> | Task Name   | $K=5$  | $K=10$ | $K=15$ | $K=20$ |
> |-------------|------|------|------|------|
> | Coffee_D0   | $54.6 \pm 6.1$ | $60.0 \pm 2$ | $59.3 \pm 7.5$  | $63.3 \pm 4.6$ |
> | StackThree_D0   | $51.3 \pm 2.3$ | $53.3 \pm 5$| $56.6 \pm 4.6$ | $54.6 \pm 5.0$  |
> | Square_D0   | $28.6 \pm 3.1$  | $31.3 \pm 3.1$ | $32.0 \pm 5.3 $ | $29.3 \pm 2.3$ |
> | ThreePieceAssembly_D0   | $39.3 \pm 5.0$ | $43.3 \pm 5.0$| $42.6 \pm 3.1$ | $34.6 \pm 4.1$ |
>
> As shown, a small $K$ may fail to capture the full failure actions, potentially causing the model to misclassify them as desirable. Larger $K$ better capture failure sequences but may introduce noise depending on the task's failure modes. Based on empirical results, we select $K=10$ as a general setting across tasks.
>
> We believe automatically identifying appropriate failure action sequences for preference learning remains an interesting direction for future work.
>
> ---
> [1] Robot Learning on the Job: Human-in-the-Loop Autonomy and Learning During Deployment. RSS 2023, Best Paper Finalist.
>
> [2] Interactive Robot Learning from Verbal Correction. CoRL 2023.
>
> [3] Model-based runtime monitoring with interactive imitation learning. ICRA 2024.

---

> > ### Comment · Reviewer_uJj6 · 2025-08-05
> >
> > Thanks for your detailed rebuttal, which addressed all of my concerns. I will raise my score.
> >
> > Best regards,
> >
> > Reviewer uJj6

---

> > > ### Author Response · Authors · 2025-08-06
> > > **Thanks for review**
> > >
> > > Dear reviewer,
> > >
> > > We are profoundly grateful for your constructive review of our work and your appreciation of our work.

---

> ### Author Response · Authors · 2025-08-04
> **Thanks for review, please let us know if you need any further clarification or additional questions.**
>
> Dear Reviewer,
>
> We sincerely thank you for your time and effort in reviewing our paper. We appreciate your feedback and suggestions.
>
> We have addressed the comments you raised in your review. Please find our responses below:
>
> 1. The experimental details of the baseline.
>
> 2. The re-implementation details of DAgger.
>
> 3. More analysis on the intrinsic mechanism of HAPO, including the proportions of different action types, ablation study of the reweighting method, and the reward curves of our method.
>
> 4. The total number of human-labeled trajectories needed for significant improvements in the real-world experiments.
>
> 5. The influence of hyper-parameters pre-intervention window size $K$ in HAPO.
>
> Please let us know if you need any further clarification or additional questions.

---

### Note · Authors · 2025-08-14

We are deeply grateful to the reviewers and the AC for their time, insightful feedback, and constructive suggestions that significantly strengthened our manuscript.

In this work, we propose to build a framework to ensure reliable continual policy optimization for large-scale VLA models, which we believe is a crucial step towards the real-world deployment of VLA models. To achieve this, we propose the Human-assisted Action Preference Optimization (HAPO) method to achieve action-level robotic behavior correction, which integrates two critical components: the human-robot collaboration framework for reliable deployment and the action preference optimization process with adaptive reweighting strategy for iterative improvement of VLA models. The experiments conducted in simulation and real-world scenarios prove superior generalization and robustness of our framework across a variety of manipulation tasks.

We thank the reviewers' positive comments on our work (address intrinsic issues in VLA optimization, important metrics for assessing a VLA optimization method, well-organized, etc.). More importantly, we also sincerely thank all the reviewers for their valuable suggestions, including more explanation of this work, more detailed analysis of the reweighting method, and statistical reporting of the improvement.

We are happy that all the reviewers' concerns were successfully addressed during the rebuttal stage. The newly added experiments and discussions will be incorporated into the revised version to further enhance the manuscript. We once again appreciate the reviewers and the AC for their insightful feedback and constructive guidance.

---

### Decision · Program_Chairs · 2025-09-17

**Decision:**

Accept (poster)

**Comment:**

The paper introduces Human-assisted Action Preference Optimization (HAPO) for continual policy optimization of Vision-Language-Action models. It initially had mixed reviews with reject but finally received leaning-positive reviews: 3 Borderline Accepts and 1 Accept.

Reviewers appreciated the strong motivation, well-structured method, and extensive experiments across both simulation and real-world robot settings. Main concerns centered on baseline details, statistical significance, and the explanation of the adaptive reweighting mechanism. During rebuttal, the authors provided additional experiments, rigorous implementation details, reward curves, and ablation studies, which help address these issues. Overall, HAPO provides a meaningful step toward reliable real-world deployment of VLA models. Given the reviewers’ updated scores, I recommend Accept.